# Unrestricted Adversarial Examples via Semantic Manipulation

**Anand Bhattad**[*] **Min Jin Chong**[*] **Kaizhao Liang** **Bo Li** **D. A. Forsyth**
University of Illinois at Urbana-Champaign
{bhattad2, mchong6, kl2, lbo, daf}@illinois.edu

## Abstract

Machine learning models, especially deep neural networks (DNNs), have been shown to be vulnerable against *adversarial examples* which are carefully crafted samples with a small magnitude of the perturbation. Such adversarial perturbations are usually restricted by bounding their $\mathcal{L}_p$ norm such that they are imperceptible, and thus many current defenses can exploit this property to reduce their adversarial impact. In this paper, we instead introduce "unrestricted" perturbations that manipulate semantically meaningful image-based visual descriptors – color and texture – in order to generate effective and photorealistic adversarial examples. We show that these semantically aware perturbations are effective against JPEG compression, feature squeezing and adversarially trained model. We also show that the proposed methods can effectively be applied to both image classification and image captioning tasks on complex datasets such as ImageNet and MSCOCO. In addition, we conduct comprehensive user studies to show that our generated semantic adversarial examples are photorealistic to humans despite large magnitude perturbations when compared to other attacks.

## 1 Introduction

Machine learning (ML), especially deep neural networks (DNNs) have achieved great success in various tasks, including image recognition (Krizhevsky et al., 2012; He et al., 2016), speech processing (Hinton et al., 2012) and robotics training (Levine et al., 2016). However, recent literature has shown that these widely deployed ML models are vulnerable to adversarial examples – carefully crafted perturbations aiming to mislead learning models (Carlini & Wagner, 2017; Kurakin et al., 2016; Xiao et al., 2018b). The fast growth of DNNs based solutions demands in-depth studies on adversarial examples to help better understand potential vulnerabilities of ML models and therefore improve their robustness.

To date, a variety of different approaches has been proposed to generate adversarial examples (Goodfellow et al., 2014b; Carlini & Wagner, 2017; Kurakin et al., 2016; Xiao et al., 2018a); and many of these attacks search for perturbation within a bounded $\mathcal{L}_p$ norm in order to preserve their photorealism. However, it is known that the $\mathcal{L}_p$ norm distance as a perceptual similarity metric is not ideal (Johnson et al., 2016; Isola et al., 2017). In addition, recent work show that defenses trained on $\mathcal{L}_p$ bounded perturbation are not robust at all against new types of unseen attacks (Kang et al., 2019). Therefore, exploring diverse adversarial examples, especially those with "unrestricted" magnitude of perturbation has acquired a lot of attention in both academia and industries (Brown et al., 2018).

Recent work based on generative adversarial networks (GANs) (Goodfellow et al., 2014a) have introduced unrestricted attacks (Song *et al*, 2018). However, these attacks are limited to datasets like MNIST, CIFAR and CelebA, and are usually unable to scale up to bigger and more complex datasets such as ImageNet. Xiao et al. (2018b) directly manipulated spatial pixel flow of an image to produce adversarial examples without $\mathcal{L}_p$ bounded constraints on the perturbation. However, the attack does not explicitly control visual semantic representation. More recently, Hosseini & Poovendran (2018) manipulated hue and saturation of an image to create adversarial perturbations. However, these examples are easily distinguishable by human and are also not scalable to complex datasets.

---

[*] indicates equal contributions.

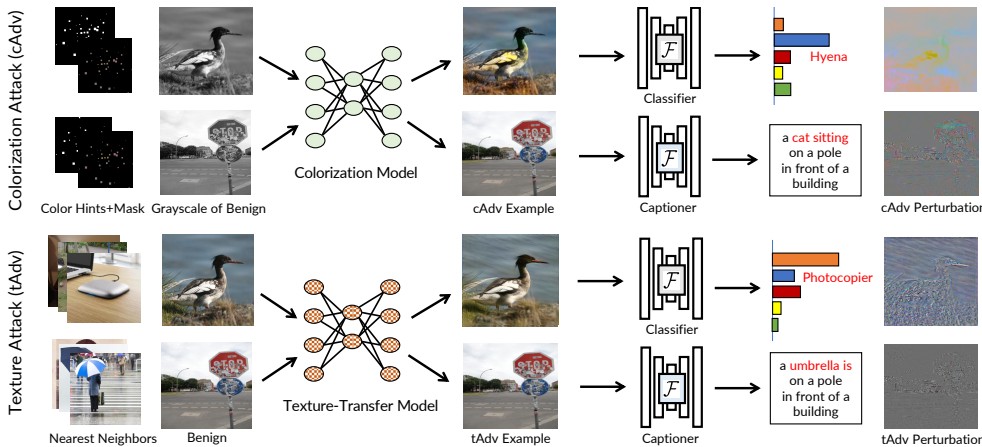

Figure 1: **An overview of proposed attacks**. **Top**: Colorization attack (**c**Adv); **Bottom**: Texture transfer attack (**t**Adv). Our attacks achieve high attack success rate via semantic manipulation without any constraints on the $\mathcal{L}_p$ norm of the perturbation. Our methods are general and can be used for attacking both classifiers and captioners.

In this work, we propose unrestricted attack strategies that explicitly manipulate semantic visual representations to generate natural-looking adversarial examples that are "far" from the original image in tems of the $\mathcal{L}_p$ norm distance. In particular, we manipulate color (**c**Adv) and texture (**t**Adv) to create realistic adversarial examples (see Fig 1). **c**Adv adaptively chooses locations in an image to change their colors, producing adversarial perturbation that is usually fairly substantial, while **t**Adv utilizes the texture from other images and adjusts the instance's texture field using style transfer.

These semantic transformation-based adversarial perturbations shed light upon the understanding of what information is important for DNNs to make predictions. For instance, in one of our case studies, when the road is recolored from gray to blue, the image gets misclassified to tench (a fish) although a car remains evidently visible (Fig. 2b). This indicates that deep learning models can easily be fooled by certain large scale patterns. In addition to image classifiers, the proposed attack methods can be generalized to different machine learning tasks such as image captioning ( Karpathy & Fei-Fei (2015)). Our attacks can either change the entire caption to the target (Chen et al., 2017; Xu et al., 2019) or take on more challenging tasks like changing one or two specific target words from the caption to a target. For example, in Fig. 1, "stop sign" of the original image caption is changed to "cat sitting" and "umbrella is" for **c**Adv and **t**Adv respectively.

To ensure our "unrestricted" semantically manipulated images are natural, we conducted extensive user studies with Amazon Mechanical Turk. We also tested our proposed attacks on several state of the art defenses. Rather than just showing the attacks break these defenses (better defenses will come up), we aim to show that **c**Adv and **t**Adv are able to produce new types of adversarial examples. Experiments also show that our proposed attacks are more transferable given their large and structured perturbations (Papernot et al., 2016). Our semantic adversarial attacks provide further insights about the vulnerabilities of ML models and therefore encourage new solutions to improve their robustness.

In summary, our **contributions** are: 1) We propose two novel approaches to generate "unrestricted" adversarial examples via semantic transformation; 2) We conduct extensive experiments to attack both image classification and image captioning models on large scale datasets (ImageNet and MSCOCO); 3) We show that our attacks are equipped with unique properties such as smooth **c**Adv perturbations and structured **t**Adv perturbations. 4) We perform comprehensive user studies to show that when compared to other attacks, our generated adversarial examples appear more natural to humans despite their large perturbations; 5) We test different adversarial examples against several state of the art defenses and show that the proposed attacks are more transferable and harder to defend.

## 2 COLORIZATION ATTACK (**c**ADV)

**Background.** Image Colorization is the task of giving natural colorization to a grayscale image. This is an ill-posed problem as there are multiple viable natural colorizations given a single grayscale

Figure 2: **Class color affinity.** Samples from unconstrained **c**Adv attacking network weights with zero hints provided. For (a) the ground truth (GT) class is `pretzel`; and for (b) the GT is `car`. These new colors added are commonly found in images from target class. For instance, green in Golfcart and blue sea in Tench images.

image. Deshpande et al. (2017) showed that diverse image colorization can be achieved by using an architecture that combines VAE (Kingma & Welling (2013)) and Mixture Density Network; while Zhang et al. (2017) demonstrated an improved and diverse image colorization by using input hints from users guided colorization process.

Our goal is to adversarially color an image by leveraging a pretrained colorization model. We hypothesize that it is possible to find a natural colorization that is adversarial for a target model (e.g., classifier or captioner) by searching in the color space. Since a colorization network learns to color natural colors that conform to boundaries and respect short-range color consistency, we can use it to introduce smooth and consistent adversarial noise with a large magnitude that looks natural to humans. This attack differs from common adversarial attacks which tend to introduce short-scale high-frequency artifacts that are minimized to be invisible for human observers.

We leverage Zhang et al. (2016; 2017) colorization model for our attack. In their work, they produce natural colorizations on ImageNet with input hints from the user. The inputs to their network consist of the L channel of the image in CIELAB color space $X_L \in \mathbb{R}^{H \times W \times 1}$, the sparse colored input hints $X_{ab} \in \mathbb{R}^{H \times W \times 2}$, and the binary mask $M \in \mathbb{B}^{H \times W \times 1}$, indicating the location of the hints.

**cAdv Objectives.** There are a few ways to leverage the colorization model to achieve adversarial objectives. We experimented with two main methods and achieved varied results.

*Network weights.* The straightforward approach of producing adversarial colors is to modify Zhang et al. (2017) colorization network, $\mathcal{C}$ directly. To do so, we simply update $\mathcal{C}$ by minimizing the adversarial loss objective $J_{adv}$, which in our case, is the cross entropy loss. $t$ represents target class and $\mathcal{F}$ represents the victim network.

$$\theta^* = \arg\min_{\theta} J_{adv}(\mathcal{F}(\mathcal{C}(X_L, X_{ab}, M; \theta)), t) \tag{1}$$

*Hints and mask.* We can also vary input hints $X_{ab}$ and mask $M$ to produce adversarial colorizations. Hints provides the network with ground truth color patches that guides the colorization, while the mask provides its spatial location. By jointly varying both hints and mask, we are able to manipulate the output colorization. We can update the hints and mask as follows:

$$M^*, X_{ab}^* = \arg\min_{M, X_{ab}} J_{adv}(\mathcal{F}(\mathcal{C}(X_L, X_{ab}, M; \theta)), t) \tag{2}$$

**cAdvAttack Methods.** Attacking network weights allows the network to search the color space with no constraints for adversarial colors. This attack is the easiest to optimize, but the output colors are not realistic as shown in Fig. 2. Our various strategies outlined below are ineffective as the model learns to generate the adversarial colors without taking into account color realism. However, colorizations produced often correlate with colors often observed in the target class. This suggests that classifiers associate certain colors with certain classes which we will discuss more in our case study.

Attacking input hints and mask jointly gives us natural results as the pretrained network will not be affected by our optimization. Attacking hints and mask separately also works but takes a long optimization time and give slightly worse results. For our experiments, we use Adam Optimizer (Kingma & Ba (2014)) with a learning rate of $10^{-4}$ in **c**Adv. We iteratively update hints and mask until our adversarial image reaches the target class and the confidence change of consecutive iterations does not exceed a threshold of $0.05$.

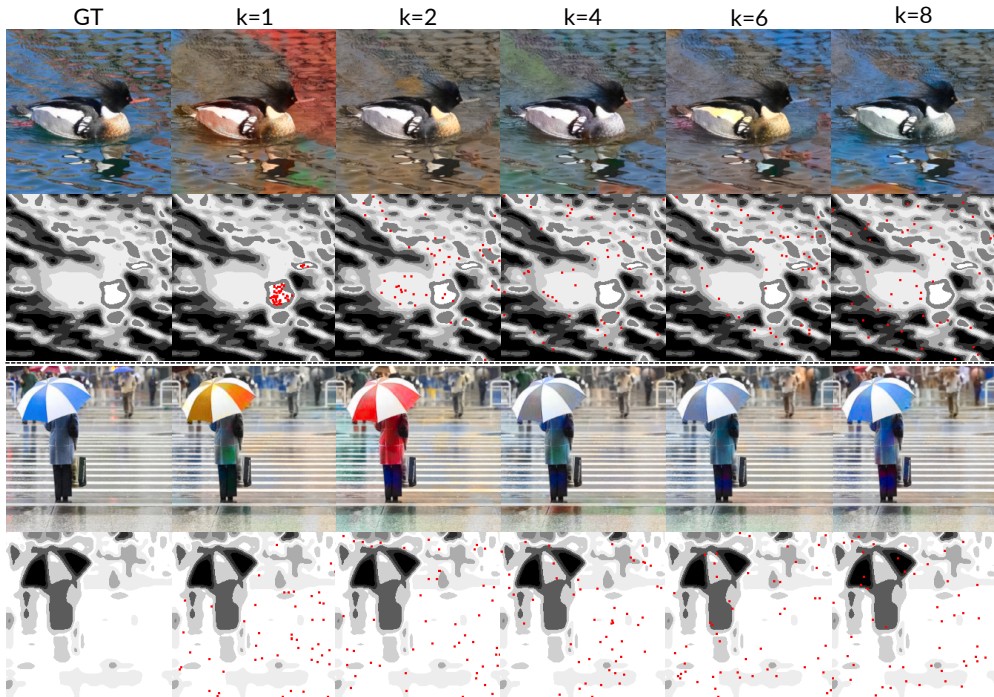

Figure 3: **Controlling cAdv**. We show a comparison of sampling 50 color hints from k clusters with low-entropy. All images are attacked to misclassify as `golf-cart`. Second and fourth row visualize our cluster segments, with darker colors representing higher mean entropy and red dots representing the sampled hints location. Sampling hints across more clusters gives less color variety.

**Control over colorization.** Current attack methods lack control over where the attack occurs, opting to attack all pixels indiscriminately. This lack of control is not important for most attacks where the $\epsilon$ is small but is concerning in cAdv where making unstructured large changes can be jarring. To produce realistic colorization, we need to avoid making large color changes at locations where colors are unambiguous (e.g. roads in general are gray) and focus on those where colors are ambiguous (e.g. an umbrella can have different colors). To do so, we need to segment an image and determine which segments should be attacked or preserved.

To segment the image into meaningful areas, we cluster the image's ground truth AB space using K-Means. We first use a Gaussian filter of $\sigma = 3$ to smooth the AB channels and then cluster them into 8 clusters. Then, we have to determine which cluster's colors should be preserved. Fortunately, Zhang et al. (2017) network output a per-pixel color distribution for a given image which we used to calculate the entropy of each pixel. The entropy represents how confident the network is at assigning a color at that location. The average entropy of each cluster represents how ambiguous their color is. We want to avoid making large changes to clusters with low-entropy while allowing our attack to change clusters with high entropy. One way to enforce this behavior is through hints, which are sampled from the ground truth at locations belonging to clusters of low-entropy. We sample hints from the $k$ clusters with the lowest entropy which we refer as $\mathbf{cAdv}_k$ (e.g. $\mathbf{cAdv}_2$ samples hints from the 2 lowest entropy clusters).

**Number of input hints.** Network hints constrain our output to have similar colors as the ground truth, avoiding the possibility of unnatural colorization at the cost of color diversity. This trade-off is controlled by the number of hints given to the network as initialization (Fig. 4). Generally, providing more hints gives us similar colors that are observed in original image. However, having too many hints is also problematic. Too many hints makes the optimization between drawing adversarial colors and matching local color hints difficult. Since the search space for adversarial colors is constrained because of more hints, we may instead generate unrealistic examples.

**Number of Clusters.** The trade-off between the color diversity and the color realism is also controlled by the number of clusters we sample hints from as shown in Fig. 3. Sampling from multiple clusters gives us realistic colors closer to the ground truth image at the expense of color diversity.

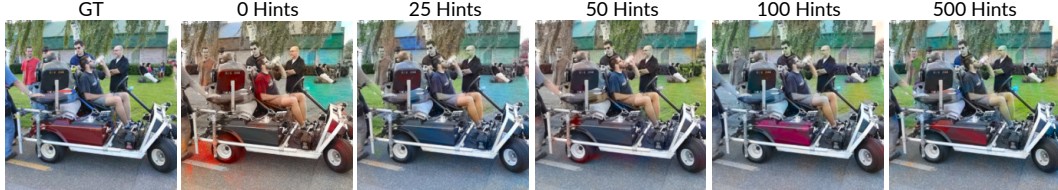

Figure 4: **Number of color hints required for cAdv**. All images are attacked to `Merganser` with $k = 4$. When the number of hints increases (from left to right), the output colors are more similar to groundtruth. However, when the number of hints is too high (500), **c**Adv often generates unrealistic perturbations. This is due to a harder optimization for **c**Adv to both add adversarial colors and match GT color hints. **c**Adv is effective and realistic with a balanced number of hints.

Empirically, from our experiments we find that in terms of color diversity, realism, and robustness of attacks, using $k = 4$ and 50 hints gives us better adversarial examples. For the rest of this paper, we fix 50 hints for all $\mathbf{cAdv}_k$ methods.

## 3 TEXTURE ATTACK (tADV)

**Background.** Texture transfer extracts texture from one image and adds it to another. Transferring texture from one source image to another target image has been widely studied in computer vision ( Efros & Freeman (2001); Gatys et al. (2015)). The Convolutional Neural Network (CNN) based texture transfer from Gatys et al. (2015) led to a series of new ideas in the domain of artistic style transfer ( Gatys et al. (2016); Huang & Belongie (2017); Li et al. (2017); Yeh et al. (2019)). More recently, Geirhos et al. (2018) showed that DNNs trained on ImageNet are biased towards texture for making predictions.

Our goal is to generate adversarial examples by infusing texture from another image without explicit constraints on $\mathcal{L}_p$ norm of the perturbation. For generating our **t**Adv examples, we used a pretrained VGG19 network (Simonyan & Zisserman, 2014) to extract textural features. We directly optimize our victim image ($I_v$) by adding texture from a target image ($I_t$). A natural strategy to transfer texture is by minimizing within-layer feature correlation statistics (gram matrices) between two images Gatys et al. (2015; 2016). Based on Yeh et al. (2019), we find that optimizing cross-layer gram matrices instead of within-layer gram matrices helps produce more natural looking adversarial examples. The difference between the within-layer and the cross-layer gram matrices is that for a within-layer, the feature's statistics are computed between the same layer. For a cross-layer, the statistics are computed between two adjacent layers.

**tAdv Objectives.** **t**Adv directly attacks the image to create adversarial examples without modifying network parameters. Moreover, there is no additional content loss that is used in style transfer methods (Gatys et al. (2016); Yeh et al. (2019)). Our overall objective function for the texture attack contains a texture transfer loss ($L_t^{\mathcal{A}}$) and an cross-entropy loss ($J_{adv}$).

$$L_{\mathbf{tAdv}}^{\mathcal{A}} = \alpha L_t^{\mathcal{A}}(I_v, I_t) + \beta J_{adv}(\mathcal{F}(I_v), t) \tag{3}$$

Unlike style transfer methods, we do not want the adversarial examples to be artistically pleasing. Our goal is to infuse a reasonable texture from a target class image to the victim image and fool a classifier or captioning network. To ensure a reasonable texture is added without overly perturbing the victim image too much, we introduce an additional constraint on the variation in the gram matrices of the victim image. This constraint helps us to control the image transformation procedure and prevents it from producing artistic images. Let $m$ and $n$ denote two layers of a pretrained VGG-19 with a decreasing spatial resolution and $C$ for number of filter maps in layer $n$, our texture transfer loss is then given by

$$L_t^{\mathcal{A}}(I_v, I_t) = \sum_{(m,n)\in\mathcal{L}} \frac{1}{C^2} \sum_{ij} \frac{\left\| G_{ij}^{m,n}(I_v) - G_{ij}^{m,n}(I_t) \right\|^2}{std\left(G_{ij}^{m,n}(I_v)\right)} \tag{4}$$

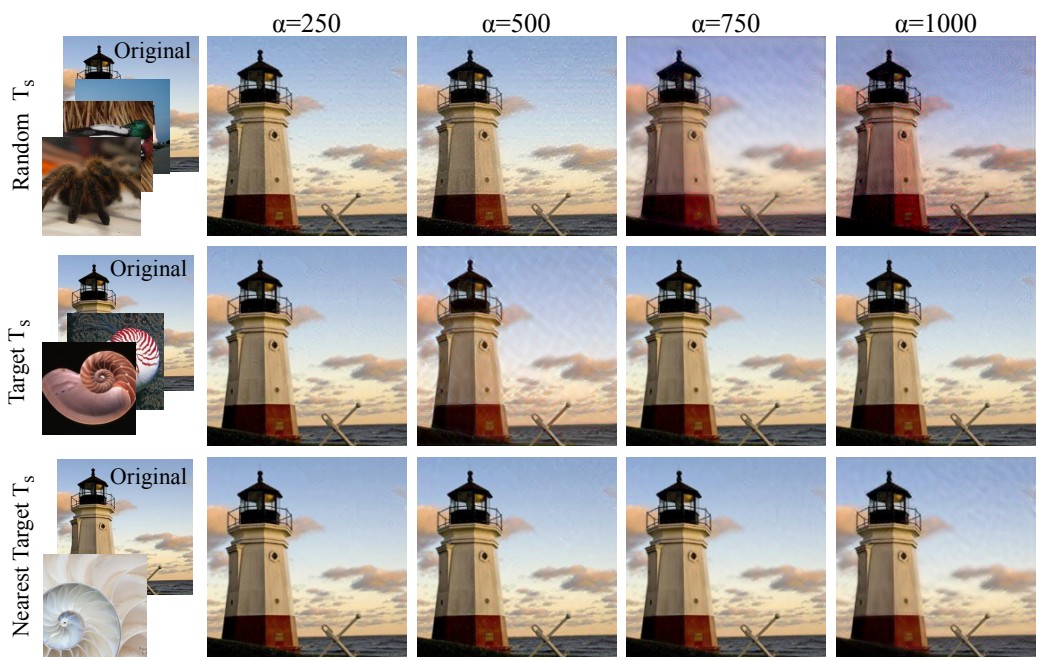

Figure 5: **tAdv strategies**. Texture transferred from random "texture source" ($T_s$) in row 1, random target class $T_s$ (row 2) and from the nearest target class $T_s$ (row 3). All examples are misclassified from `Beacon` to `Nautilus`. Images in the last row look photo realistic, while those in the first two rows contain more artifacts as the texture weight $\alpha$ increases (left to right).

Let $f$ be feature maps, $\mathcal{U}f^n$ be an upsampled $f^n$ that matches the spatial resolution of layer $m$. The cross layer gram matrices $G$ between the victim image ($I_v$) and a target image ($I_t$) is given as

$$G_{ij}^{m,n}(I) = \sum_p \left[ f_{i,p}^m(I) \right] \left[ \mathcal{U}f_{j,p}^n(I) \right]^T \tag{5}$$

**Texture Transfer.** To create **t**Adv adversarial examples, we need to find images to extract the texture from, which we call "texture source" ($T_s$). A naive strategy is to randomly select an image from the data bank as $T_s$. Though this strategy is successful, their perturbations are clearly perceptible. Alternatively, we can randomly select $T_s$ from the adversarial target class. This strategy produces less perceptible perturbations compared to the random $T_s$ method as we are extracting a texture from the known target class. A better strategy to select $T_s$ is to find a target class image that lies closest to the victim image in the feature space using nearest neighbors. This strategy is sensible as we assure our victim image has similar feature statistics as our target image. Consequently, minimizing gram matrices is easier and our attack generates more natural looking images (see Fig. 5).

For texture transfer, we extract cross-layer statistics in Eq. 4 from the R11, R21, R31, R41, and R51 of a pretrained VGG19. We optimize our objective (Eq. 3) using an L-BFGS (Liu & Nocedal (1989)) optimizer. **t**Adv attacks are sensitive and if not controlled well, images get transformed into artistic images. Since we do not have any constraints over the perturbation norm, it is necessary to decide when to stop the texture transfer procedure. For a successful attack (images look realistic), we limit our L-BFGS to fixed number of small steps and perform two set of experiments: one with only one iteration or round of L-BFGS for 14 steps and another with three iterations of 14 steps. For the three iterations setup, after every iteration, we look at the confidence of our target class and stop if the confidence is greater than 0.9.

**Texture and Cross-Entropy Weights.** Empirically, we found setting $\alpha$ to be in the range $[150, 1000]$ and $\beta$ in the range $\left[ 10^{-4}, 10^{-3} \right]$ to be successful and also produce less perceptible **t**Adv examples. The additional cross-entropy based adversarial objective $J_{adv}$ helps our optimization. We ensure large flow of gradients is from the texture loss and they are sufficiently larger than the adversarial cross-entropy objective. The adversarial objective also helps in transforming victim image to adversarial without stylizing the image. All our tabulated results are shown for one iteration, $\alpha = 250$ and $\beta = 10^{-3}$, unless otherwise stated. We use the annotation **t**Adv$_\alpha^{iter}$ for the rest of the paper to denote the texture method that we are using.

| | Model | R50 | D121 | VGG19 |
|---|---|---|---|---|
| | Accuracy | 76.15 | 74.65 | 74.24 |
| Attack Success | $\mathbf{c}Adv_1$ | 99.72 | 99.89 | 99.89 |
| | $\mathbf{c}Adv_4$ | 99.78 | 99.83 | 100.00 |
| | $\mathbf{t}Adv_{250}^1$ | 97.99 | 99.72 | 99.50 |
| | $\mathbf{t}Adv_{500}^1$ | 99.27 | 99.83 | 99.83 |

**(a) Whitebox Target Attack Success Rate**

| Method | Model | R50 | D121 | VGG19 |
|---|---|---|---|---|
| Kurakin et al. (2016) | R50 | 100.00 | 17.33 | 12.95 |
| Carlini & Wagner (2017) | R50 | 98.85 | 16.50 | 11.00 |
| Xiao et al. (2018b) | R50 | 100 | 5.23 | 8.90 |
| $\mathbf{c}Adv_4$ | R50 | 99.83 | 28.56 | 31.00 |
| | D121 | 18.13 | 99.83 | 29.43 |
| | VGG19 | 22.94 | 26.39 | 100.00 |
| $\mathbf{t}Adv_{250}^1$ | R50 | 99.00 | 24.51 | 34.84 |
| | D121 | 21.16 | 99.83 | 32.56 |
| | VGG19 | 20.21 | 24.40 | 99.89 |

**(b) Transferability**

Table 1: Our attacks are highly successful on ResNet50 (R50), DenseNet121 (D121) and VGG19. In (a), for $\mathbf{c}$Adv, we show results for $k = \{1, 4\}$ when attacked with 50 hints. For $\mathbf{t}$Adv we show results for $\alpha = \{250, 500\}$ and $\beta = 0.001$. In (b) We show the transferability of our attacks. We attack models from the column and test them on models from the rows.

**Control over Texture.** The amount of texture that gets added to our victim image is controlled by the texture weight coefficient ($\alpha$). Increasing texture weights improves attack success rate at the cost of noticeable perturbation. When compared to within-layer statistics, the cross-layer statistics that we use are not only better at extracting texture, it is also easier to control the texture weight.

## 4 EXPERIMENTAL RESULTS

In this section, we evaluate the two proposed attack methods both quantitatively, via attack success rate under different settings, and qualitatively, based on interesting case studies. We conduct our experiments on ImageNet Deng et al. (2009) by randomly selecting images from 10 sufficiently different classes predicted correctly for the classification attack.

We use a pretrained ResNet 50 classifier (He et al. (2016)) for all our methods. DenseNet 121 and VGG 19 (Huang et al.; Simonyan & Zisserman (2014)) are used for our transferability analysis.

### 4.1 $\mathbf{c}$ADV ATTACK

$\mathbf{c}$Adv achieves high targeted attack success rate by adding realistic color perturbation. Our numbers in Table 1 and Table 2 also reveal that $\mathbf{c}$Adv examples with larger color changes (consequently more color diversity) are more robust against transferability and adversarial defenses. However, these big changes are found to be slightly less realistic from our user study (Table 2, Table 4).

**Smooth $\mathbf{c}$Adv perturbations.** Fig. 8 in our Appendix shows interesting properties of the adversarial colors. We observe that $\mathbf{c}$Adv perturbations are locally smooth and are relatively low-frequency. This is different from most adversarial attacks that generate high-frequency noise-like perturbations. This phenomenon can be explained by the observation that colors are usually smooth within object boundaries. The pretrained colorization model will thus produce smooth, low-frequency adversarial colors that conform to object boundaries.

**Importance of color in classification**. From Fig. 2, we can compare how different target class affects our colorization results if we relax our constraints on colors ($\mathbf{c}$Adv on Network Weights, 0 hints). In many cases, the images contain strong colors that are related to the target class. In the case of golf-cart, we get a green tint over the entire image. This can push the target classifier to misclassify the image as green grass is usually overabundant in benign golf-cart images. Fig. 2b shows our attack on an image of a car to tench (a type of fish). We observe that the gray road turned blue and that the colors are tinted. We can hypothesize that the blue colors and the tint fooled the classifier into thinking the image is a tench in the sea.

The colorization model is originally trained to produce natural colorization that conforms to object boundaries. By adjusting its parameters, we are able to produce such large and abnormal color change that is impossible with our attack on hints and mask. These colors, however, show us some evidence that colors play a stronger role in classification than we thought. We reserve the exploration of this observation for future works.

While this effect (strong color correlation to target class) is less pronounced for our attack on hints and mask, for all $\mathbf{c}$Adv methods, we observe isoluminant color blobs. Isoluminant colors are characterized

| Method | Res50 | JPEG75 | Feature Squeezing | | | | | Res152 | Adv Res152 | User Pref. |
|---|---|---|---|---|---|---|---|---|---|---|
| | | | 4-bit | 5-bit | 2x2 | 3x3 | 11-3-4 | | | |
| Kurakin et al. (2016) | 100 | 12.73 | 28.62 | 86.66 | 34.28 | 21.56 | 29.28 | 1.08 | 1.38 | 0.506 |
| Carlini & Wagner (2017) | 99.85 | 11.50 | 12.00 | 30.50 | 22.00 | 14.50 | 18.50 | 1.08 | 1.38 | 0.497 |
| Hosseini & Poovendran (2018) | 1.20 | – | – | – | – | – | – | – | – | – |
| Xiao et al. (2018b) | 100 | 17.61 | 22.51 | 29.26 | 28.71 | 23.51 | 26.67 | 4.13 | 1.39 | 0.470 |
| $cAdv_1$ | 100 | **52.33** | **47.78** | **76.17** | **36.28** | **50.50** | **61.95** | 12.06 | 11.62 | 0.427 |
| $cAdv_2$ | 99.89 | 46.61 | 42.78 | 72.56 | 34.28 | 46.45 | 59.00 | **17.39** | **19.4** | 0.437 |
| $cAdv_4$ | 99.83 | 42.61 | 38.39 | 69.67 | 34.34 | 40.78 | 54.62 | 14.13 | 12.5 | 0.473 |
| $cAdv_8$ | 99.81 | 38.22 | 36.62 | 67.06 | 31.67 | 37.67 | 49.17 | 6.52 | 10.04 | **0.476** |
| $tAdv_{250}^1$ | 99.00 | 32.89 | 62.79 | 89.74 | 54.94 | 38.92 | 40.57 | 10.9 | 2.10 | 0.433 |
| $tAdv_{250}^3$ | 100 | **36.33** | **67.68** | **94.11** | **58.92** | **42.82** | 44.56 | 15.21 | 4.6 | 0.425 |
| $tAdv_{1000}^1$ | 99.88 | 31.49 | 52.69 | 90.52 | 51.24 | 34.85 | 39.68 | 19.12 | 5.59 | 0.412 |
| $tAdv_{1000}^3$ | 100 | 35.23 | 61.40 | 93.18 | 56.31 | 39.66 | **45.59** | **22.28** | **6.94** | 0.406 |

Table 2: **Comparison against defense models**. Misclassification rate after passing different adversarial examples through the defense models (higher means the attack is stronger). All attacks are performed on ResNet50 with whitebox attack. The highest attack success rate is in bold. We also report the user preference scores from AMT for each attack (last column).

by a change in color without a corresponding change in luminance. As most color changes occur along edges in natural images, it is likely that classifiers trained on ImageNet have never seen isoluminant colors. This suggests that **c**Adv might be exploiting isoluminant colors to fool classifiers.

## 4.2 tADV ATTACK

**t**Adv successfully fools the classifiers with a very small weighted adversarial cross-entropy objective ($\beta$) when combined with texture loss, while remaining realistic to humans. As shown in Table 1, our attacks are highly successful on white-box attacks tested on three different models with the nearest neighbor texture transfer approach. We also show our attacks are more transferable to other models. In our Appendix, we show ablation results for **t**Adv attacks along with other strategies that we used for generating **t**Adv adversarial examples.

**Structured tAdv Perturbations.** Since we extract features across different layers of VGG, the **t**Adv perturbations follow a textural pattern. They are more structured and organized when compared to others. Our **t**Adv perturbations are big when compared with existing attack methods in $\mathcal{L}_p$ norm. They are of high-frequency and yet imperceptible (see Fig. 1 and Fig. 8).

**Importance of Texture in Classification.** Textures are crucial descriptors for image classification and Imagenet trained models can be exploited by altering the texture. Their importance is also shown in the recent work from Geirhos et al. (2018). Our results also shows that even with a small or invisible change in the texture field can break the current state of the art classifiers.

## 4.3 DEFENSE AND TRANSFERABILITY ANALYSIS

We test all our attacks and other existing methods with images attacked from Resnet50. We evaluate them on three defenses – JPEG defense (Das et al., 2017), feature squeezing (Xu et al., 2017) and adversarial training. By leveraging JPEG compression and decompression, adversarial noise may be removed. We tested our methods against JPEG compression of 75. Feature squeezing is a family of simple but surprisingly effective strategies, including reducing color bit depth and spatial smoothing. Adversarial training has been shown as an effective but costly method to defend against adversarial attacks. Mixing adversarial samples into training data of a classifier improves its robustness without affecting the overall accuracy. We were able to obtain an adversarially pretrained Resnet152 model on ImageNet dataset and hence we tested our Resnet50 attacked images with this model.

**Robustness.** In general, our attacks are more robust to the considered defenses and transferable for targeted attacks. For **c**Adv, there is a trade-off between more realistic colors (using more hints and sampling from more clusters) and attack robustness. From Table 1 and 2, we show that as we progressively use more clusters, our transferability and defense numbers drop. A similar trend is observed with the change in the number of hints. **c**Adv is robust to JPEG defense and adversarial training because of their large and spatially smooth perturbations. For **t**Adv, increasing texture weight ($\alpha$) does not necessarily perform well with the defense even though it increases attack success rate, but increasing texture flow with more iterations improves attack's robustness against defenses.

## 5 HUMAN PERCEPTUAL STUDIES

To quantify how realistic **t**Adv and **c**Adv examples are, we conducted a user study on Amazon Mechanical Turk (AMT). We follow the same procedure as described in (Zhang et al., 2016; Xiao et al., 2018b). For each attack, we choose the same 200 adversarial images and their corresponding benign ones. During each trial, one random adversarial-benign pair appears for three seconds and workers are given five minutes to identify the realistic one. Each attack has 600 unique pairs of images and each pair is evaluated by at least 10 unique workers. We restrict biases in this process by allowing each unique user up to 5 rounds of trials and also ignore users who complete the study in less than 30 seconds. In total, 598 unique workers completed at least one round of our user study. For each image, we can then calculate the user preference score as the number of times it is chosen divided by the number of times it is displayed. $0.5$ represents that users are unable to distinguish if the image is fake. For **c**Adv and **t**Adv, user preferences averages at $0.476$ and $0.433$ respectively, indicating that workers have a hard time distinguishing them. The user preferences for all attacks are summarized in Table 2 and their comparison with $\mathcal{L}_p$ norm is in Table 4 and Table 5.

## 6 ATTACKING CAPTIONING MODEL

Our methods are general and can be easily adapted for other learning tasks. As proof of concept, we test our attacks against image captioning task. **Image captioning** is the task of generating a sequence of word description for an image. The popular architecture for captioning is a Long-Short-Term-Memory (LSTM) (Hochreiter & Schmidhuber, 1997) based models (Karpathy & Fei-Fei, 2015; Wang et al., 2017). Recently, (Aneja et al., 2018) proposed a convolutional based captioning model for a fast and accurate caption generation. This convolutional based approach does not suffer from the commonly known problems of vanishing gradients and overly confident predictions of LSTM network. Therefore, we choose to attack the current state of the art convolutional captioning model. We randomly selected images from MSCOCO (Lin et al., 2014) for image captioning attack.

Attacking captioning models is harder than attacking classifiers when the goal is to change exactly one word in the benign image's caption unlike pixel based attacks (Chen et al., 2017; Xu et al., 2019). We show that our attacks are successful and have no visible artifacts even for this challenging task. In Fig. 6, we change the second word of the caption to *dog* while keeping the rest of the caption the same. This is a challenging targeted attack because, in many untargeted attacks, the resulted captions do not make sense. More examples are in our Appendix.

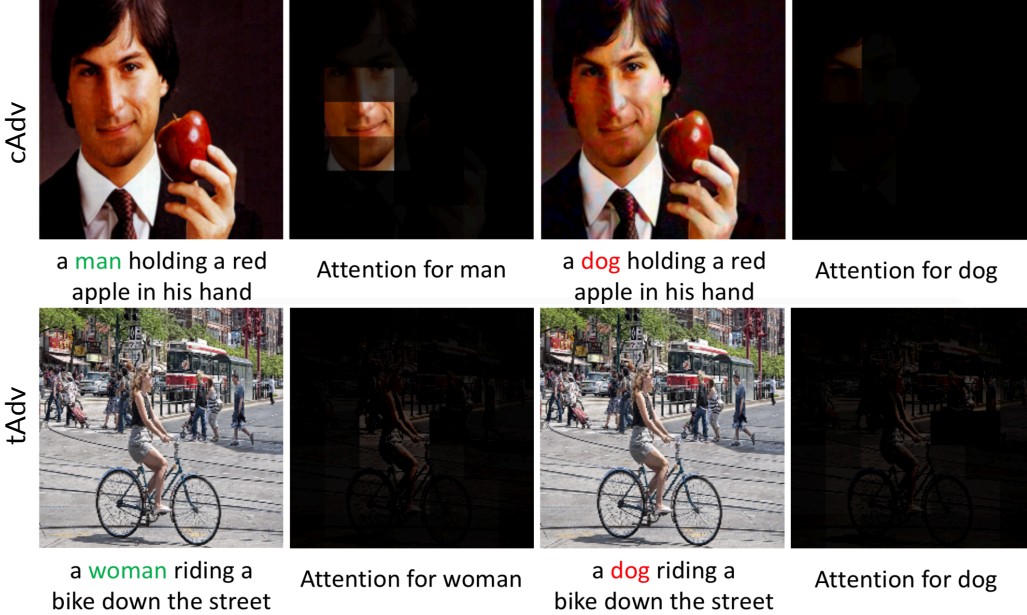

Figure 6: **Captioning attack**. **Top**: **c**Adv; **Bottom**: **t**Adv. We attack the second word to dog and show the corresponding change in attention mask of that word. More examples in Appendix.

**Adversarial Cross-Entropy Objective for Captioning.** Let $t$ be the target caption, $w$ denote the word position of the caption, $\mathcal{F}$ for the captioning model, $I_v$ for the victim image and $J_{adv}$ for the cross-entropy loss

$$L_{capt}^{\mathcal{A}} = \sum_w J_{adv}((\mathcal{F}(I_v))_w, t_w) \tag{6}$$

For **c**Adv, we give all color hints and optimize to get an adversarial colored image to produce target caption. For **t**Adv, we add Eqn 6 to Eqn 4 to optimize the image. We select $T_S$ as the nearest neighbor of the victim image from the ones in the adversarial target class using ImageNet dataset. We stop our attack once we reach the target caption and the caption does not change in consecutive iterations. Note we do not change the network weights, we only optimize hints and mask (for **c**Adv) or the victim image (for **t**Adv) to achieve our target caption.

## 7 RELATED WORK

Here we briefly summarize existing unrestricted and semantic adversarial attacks. Xiao et al. (2018b) proposed geometric or spatial distortion of pixels in image to create adversarial examples. They distort the input image by optimizing pixel flow instead of pixel values to generate adversarial examples. While this attack leads to "natural" looking adversarial examples with large $\mathcal{L}_\infty$ norm, it does not take image semantics into account. Song *et al* (2018) and Dunn et al. (2019) considered GANs for adversarial attacks. This attack is unrestricted in $\mathcal{L}_p$ norm but they are restricted to simple datasets as it involves training GANs, which have been known to be unstable and computationally intensive for complex datasets like ImageNet (Karras et al., 2017; Brock et al., 2018).

Hosseini & Poovendran (2018), changes the hue & saturation of an image randomly to create adversarial examples. It is similar to **c**Adv as they both involve changing colors, however, their search space is limited to two dimensions and their images are unrealistic, Appendix (Fig. 10). Also, while this method has a non-trivial untargeted attack success rate, it performs extremely poorly for targeted attacks (1.20% success rate in our own experiments on ImageNet). Our work is also related to Joshi et al. (2019) and Qiu et al. (2019), who manipulate images conditioned on face dataset attributes like glasses, beard for their attacks. These work focuses on changing single image visual attribute and are conditionally dependent. Our work focuses on changing visual semantic descriptors to misclassify images and are not conditioned to any semantic attributes.

| Method | Semantic Based | Unrestricted | Photorealistic | Explainable (eg color affinity) | Complex Dataset | Caption Attack |
|---|---|---|---|---|---|---|
| Kurakin et al. (2016) | ✗ | ✗ | ✓ | ✗ | ✓ | ✗ |
| Carlini & Wagner (2017) | ✗ | ✗ | ✓ | ✗ | ✓ | ✗ |
| Hosseini & Poovendran (2018) | ✓ | ✓ | ✗ | ✗ | ✗ | ✗ |
| Song *et al* (2018) | ✗ | ✓ | ✗ | ✗ | ✗ | ✗ |
| Xiao et al. (2018b) | ✗ | ✓ | ✓ | ✓ | ✓ | ✗ |
| **c**Adv (**ours**) | ✓ | ✓ | ✓ | ✓ | ✓ | ✓ |
| **t**Adv (**ours**) | ✓ | ✓ | ✓ | ✓ | ✓ | ✓ |

Table 3: Summary of the difference in our work compared to previous work. Unlike previous attack methods, our attacks are unbounded, semantically motivated, realistic, highly successful, and scales to more complex datasets and other ML tasks. They are also robust against tested defenses.

## 8 CONCLUSION

Our proposed two novel unrestricted semantic attacks shed light on the role of texture and color fields in influencing DNN's predictions. They not only consistently fool human subjects but in general are harder to defend against. We hope by presenting our methods, we encourage future studies on unbounded adversarial attacks, better metrics for measuring perturbations, and more sophisticated defenses.

### ACKNOWLEDGEMENTS

We thank Chaowei Xiao for sharing their code to compare our methods with Xiao et al. (2018b) and helping us setup the user study. We also thank Tianyuan Zhang for providing the AdvRes152 pretrained model. This work was supported by NSF Grant No. 1718221 and ONR MURI Award N00014-16-1-2007.

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

# A APPENDIX

## A.1 OTHER DETAILS ON HUMAN STUDY

We also chose BIM (Kurakin et al., 2016) and CW (Carlini & Wagner, 2017) for comparing our perturbations. Since these attacks are known to have low $\mathcal{L}_p$ norm, we designed an aggressive version of BIM by relaxing its $\mathcal{L}_\infty$ bound to match the norm of our attacks. We settled with two aggressive versions of BIM with average $\mathcal{L}_\infty = \{0.21, 0.347\}$, which we refer to as $\text{BIM}_{0.21}$, $\text{BIM}_{0.34}$. The average user preferences for BIM drops drastically from 0.497 to 0.332 when we relax the norm to $\text{BIM}_{0.34}$; the decrease in user preferences for tAdv (0.433 to 0.406) and cAdv (0.476 to 0.437) is not significant. In Fig. 7, we plot a density plot of $\mathcal{L}_\infty$ vs user preference scores.

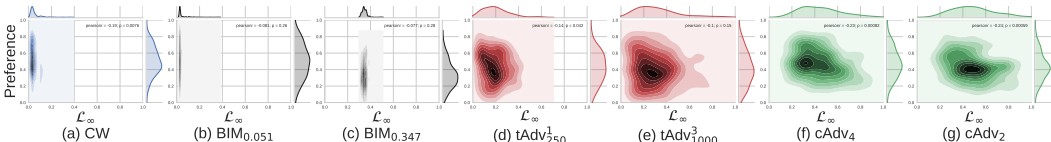

Figure 7: **Density Plot.** Our methods achieve large $\mathcal{L}_\infty$ norm perturbations without notable reduction in user preference. Each plot is a density plot between perturbation ($\mathcal{L}_\infty$ norm) on X axis and $Pr$(user prefers adversarial image) on Y axis. For ideal systems, the density would be a concentrated horizontal line at 0.5. All plots are on the same set of axes. On the **left**, plots for three baseline methods (Fig a - Fig c). Note the very strong concentration on small norm perturbations, which users like. **Right** 4 plots shows our methods (Fig d - Fig g). Note strong push into large norm regions, without loss of user preference.

| Method | $\mathcal{L}_0$ | $\mathcal{L}_2$ | $\mathcal{L}_\infty$ | Preference |
|---|---|---|---|---|
| cw | 0.587 | 0.026 | 0.054 | 0.506 |
| $\text{BIM}_{0.051}$ | 0.826 | 0.030 | 0.051 | 0.497 |
| $\text{BIM}_{0.21}$ | 0.893 | 0.118 | 0.21 | 0.355 |
| $\text{BIM}_{0.347}$ | 0.892 | 0.119 | 0.347 | 0.332 |
| $\text{cAdv}_2$ | 0.910 | 0.098 | 0.489 | 0.437 |
| $\text{cAdv}_4$ | 0.898 | 0.071 | 0.448 | 0.473 |
| $\text{cAdv}_8$ | 0.896 | 0.059 | 0.425 | 0.476 |
| $\text{tAdv}_{250}^1$ | 0.891 | 0.041 | 0.198 | 0.433 |
| $\text{tAdv}_{1000}^1$ | 0.931 | 0.056 | 0.237 | 0.412 |
| $\text{tAdv}_{250}^3$ | 0.916 | 0.061 | 0.258 | 0.425 |
| $\text{tAdv}_{1000}^3$ | 0.946 | 0.08 | 0.315 | 0.406 |

Table 4: **User Study.** User preference score and $\mathcal{L}_p$ norm of perturbation for different attacks. cAdv and tAdv are imperceptible for humans (score close to 0.5) even with very big $\mathcal{L}_p$ norm perturbation.

| Attack | | tarantula | merganser | nautilus | hyena | beacon | golfcart | photocopier | umbrella | pretzel | sandbar | Mean |
|---|---|---|---|---|---|---|---|---|---|---|---|---|
| BIM | $\mathcal{L}_0$ | 0.82 | 0.81 | 0.80 | 0.88 | 0.80 | 0.79 | 0.86 | 0.80 | 0.85 | 0.84 | 0.83 |
| | $\mathcal{L}_2$ | 0.05 | 0.02 | 0.03 | 0.04 | 0.02 | 0.01 | 0.04 | 0.02 | 0.02 | 0.04 | 0.03 |
| | $\mathcal{L}_\infty$ | 0.07 | 0.04 | 0.06 | 0.06 | 0.04 | 0.03 | 0.07 | 0.04 | 0.04 | 0.06 | 0.05 |
| | Preference | 0.49 | 0.75 | 0.44 | 0.67 | 0.48 | 0.48 | 0.27 | 0.39 | 0.38 | 0.63 | 0.50 |
| CW | $\mathcal{L}_0$ | 0.54 | 0.55 | 0.52 | 0.77 | 0.48 | 0.42 | 0.70 | 0.48 | 0.69 | 0.71 | 0.59 |
| | $\mathcal{L}_2$ | 0.05 | 0.01 | 0.03 | 0.03 | 0.02 | 0.01 | 0.04 | 0.01 | 0.02 | 0.04 | 0.03 |
| | $\mathcal{L}_\infty$ | 0.08 | 0.04 | 0.06 | 0.07 | 0.04 | 0.03 | 0.07 | 0.04 | 0.05 | 0.06 | 0.05 |
| | Preference | 0.51 | 0.68 | 0.43 | 0.59 | 0.52 | 0.51 | 0.39 | 0.40 | 0.42 | 0.62 | 0.51 |
| $\text{tAdv}_{250}^3$ | $\mathcal{L}_0$ | 0.94 | 0.95 | 0.91 | 0.94 | 0.89 | 0.94 | 0.89 | 0.91 | 0.94 | 0.85 | 0.92 |
| | $\mathcal{L}_2$ | 0.07 | 0.08 | 0.08 | 0.06 | 0.05 | 0.06 | 0.03 | 0.05 | 0.07 | 0.05 | 0.06 |
| | $\mathcal{L}_\infty$ | 0.28 | 0.31 | 0.30 | 0.31 | 0.22 | 0.27 | 0.18 | 0.25 | 0.27 | 0.19 | 0.26 |
| | Preference | 0.39 | 0.58 | 0.43 | 0.53 | 0.46 | 0.35 | 0.25 | 0.40 | 0.33 | 0.54 | 0.43 |
| $\text{cAdv}_4$ | $\mathcal{L}_0$ | 0.90 | 0.90 | 0.88 | 0.90 | 0.90 | 0.90 | 0.90 | 0.89 | 0.93 | 0.88 | 0.90 |
| | $\mathcal{L}_2$ | 0.06 | 0.06 | 0.07 | 0.05 | 0.08 | 0.07 | 0.08 | 0.08 | 0.09 | 0.06 | 0.07 |
| | $\mathcal{L}_\infty$ | 0.36 | 0.44 | 0.41 | 0.31 | 0.48 | 0.51 | 0.55 | 0.56 | 0.46 | 0.39 | 0.45 |
| | Preference | 0.46 | 0.65 | 0.46 | 0.58 | 0.45 | 0.48 | 0.30 | 0.43 | 0.35 | 0.59 | 0.47 |

Table 5: Class wise $\mathcal{L}_p$ norm and user preference breakdown. Users are biased and pick a few classes quite often (`merganser`, `sandbar`), and do not like a few classes (`photocopier`) over others.

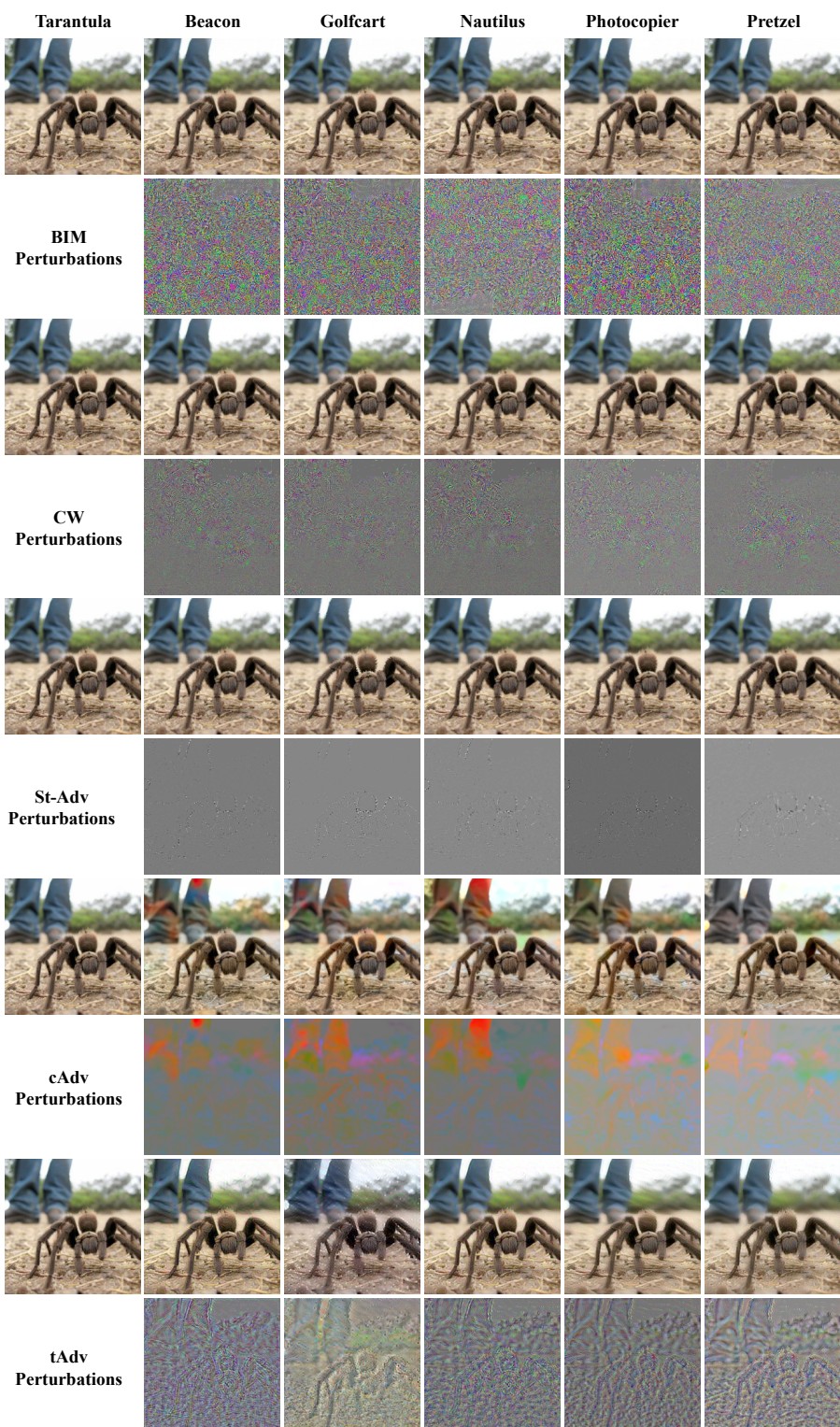

Figure 8: **Perturbation comparisons**. Images are attacked from `tarantula` to `beacon`, `golf cart`, `nautilus`, `photocopier`, `pretzel` from left to right. Our perturbations (**c**Adv and **t**Adv) are large, structured and have spatial patterns when compared with other attacks. Perturbations from **c**Adv are low-frequency and locally smooth while perturbations from **t**Adv are primarily high-frequency and structured. Note gray color indicates no perturbations.

## A.2 ADDITIONAL RESULTS

| | Model | Resnet50 | Dense121 | VGG 19 |
|---|---|---|---|---|
| | Accuracy | 76.15 | 74.65 | 74.24 |
| | Random $T_s$ | 99.67 | 99.72 | 96.16 |
| | Random Target $T_s$ | 99.72 | 99.89 | 99.94 |
| Attack Success | Nearest Target $T_s$ | 97.99 | 99.72 | 99.50 |
| | cAdv$_4$ 25 hints | 99.78 | 99.83 | 99.93 |
| | cAdv$_4$ 50 hints | 99.78 | 99.83 | 100.00 |
| | cAdv$_4$ 100 hints | 99.44 | 99.50 | 99.93 |

**Whitebox target attack success rate.** Our attacks are highly successful on different models across all strategies. tAdv results are for $\alpha = 250$, $\beta = 10^{-3}$ and iter$= 1$.

| $\beta$ \ $\alpha$ | 250 | 500 | 750 | 1000 |
|---|---|---|---|---|
| 0 | 25.00 | 99.61 | 98.55 | 95.92 |
| $10^{-4}$ | 99.88 | 99.61 | 98.55 | 95.92 |
| $10^{-3}$ | 97.99 | 99.27 | 99.66 | 99.50 |
| $10^{-2}$ | 96.26 | 95.42 | 96.32 | 96.59 |

**tAdv ablation study.** Whitebox target success rate with nearest target $T_s$ (texture source). In columns, we have increasing texture weight ($\alpha$) and in rows, we have increasing adversarial cross-entropy weight ($\beta$). All attacks are done on Resnet50.

Table 6: Ablation Studies.

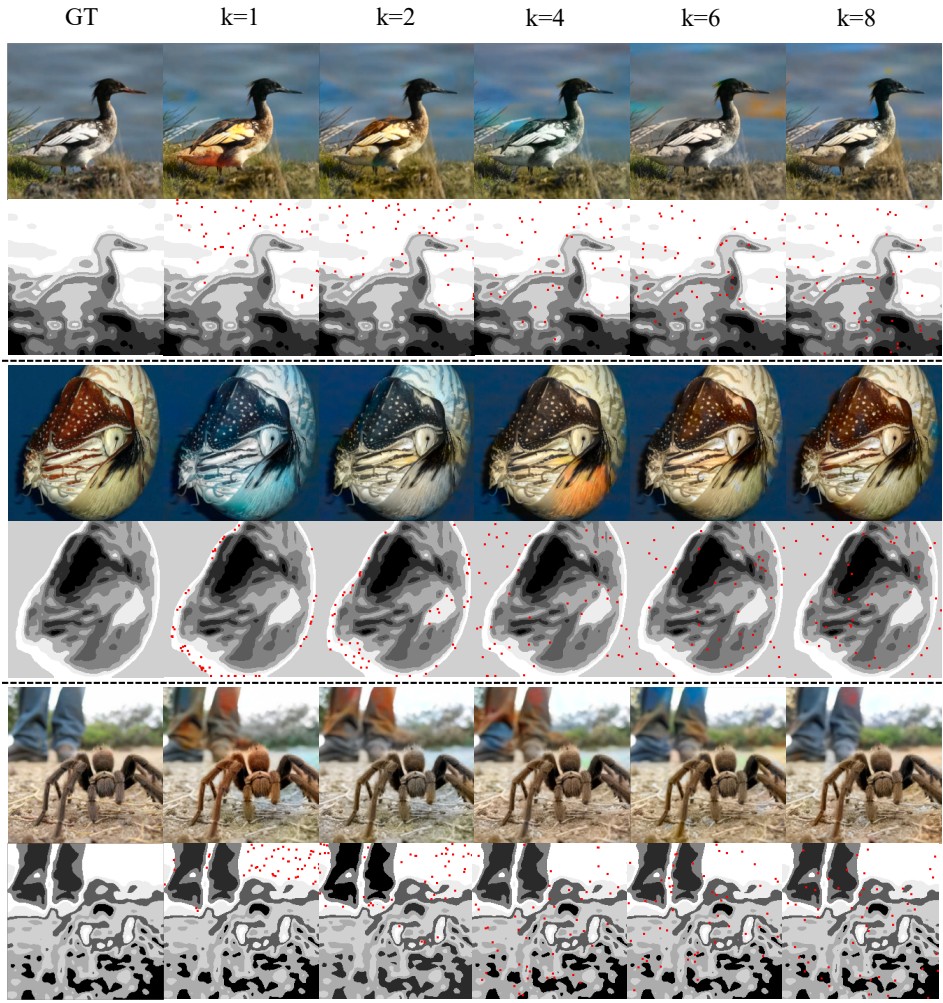

Figure 9: **Additional qualitative examples for controlling cAdv**. We show a comparison of sampling 50 color hints from k clusters with low-entropy. All images are attacked to `golf-cart`. Even numbered rows visualize our cluster segments, with darker colors representing higher mean entropy and red dots representing the location we sample hints from. Sampling hints across more clusters gives less color variety.

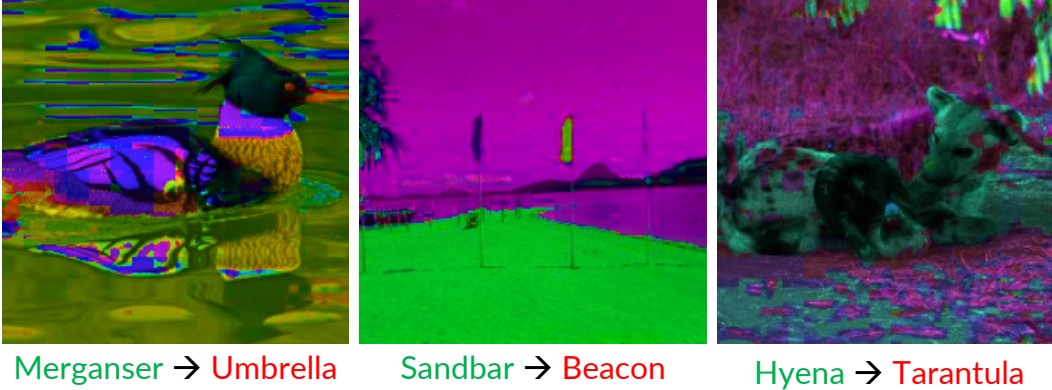

Merganser → Umbrella  Sandbar → Beacon  Hyena → Tarantula

Figure 10: Images attacked with the method of (Hosseini & Poovendran, 2018) are not realistic, as these examples show. Compare Fig 12, showing results of our color attack. Similar qualitative results for CIFAR-10 are visible in Hosseini & Poovendran (2018).

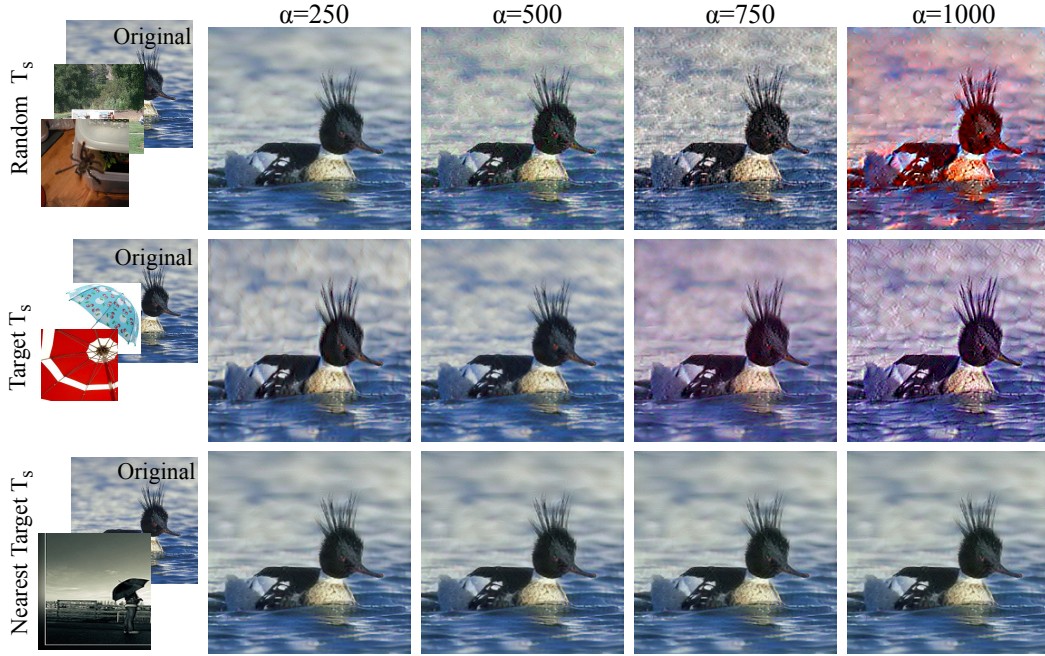

Figure 11: **Additional qualitative examples for tAdv**. Texture transferred from random images (row 1), random images from adversarial target class (row 2) and from the nearest neighbor of the victim image from the adversarial target class (row 3). All examples are misclassified from Merganser to Umbrella. Images in the last row look photo realistic, while those in the first two rows contain more artifacts as the texture weight $\alpha$ increases (left to right).

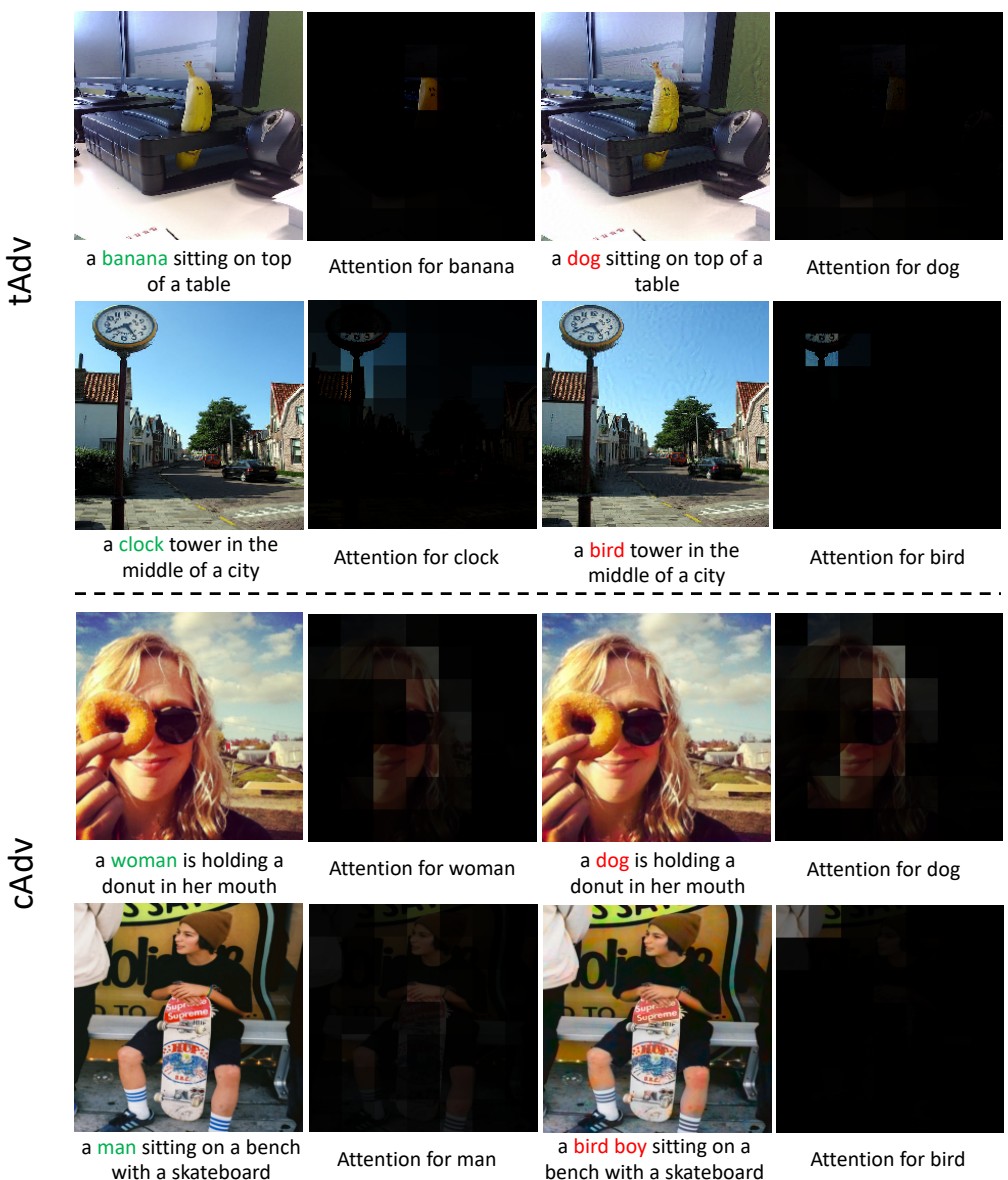

Figure 12: **Captioning attack**. We attack the second word of each caption to {dog, bird} and show the corresponding change in attention mask of that word. For tAdv we use the nearest neighbor selection method and for cAdv we initialize with all groundtruth color hints.

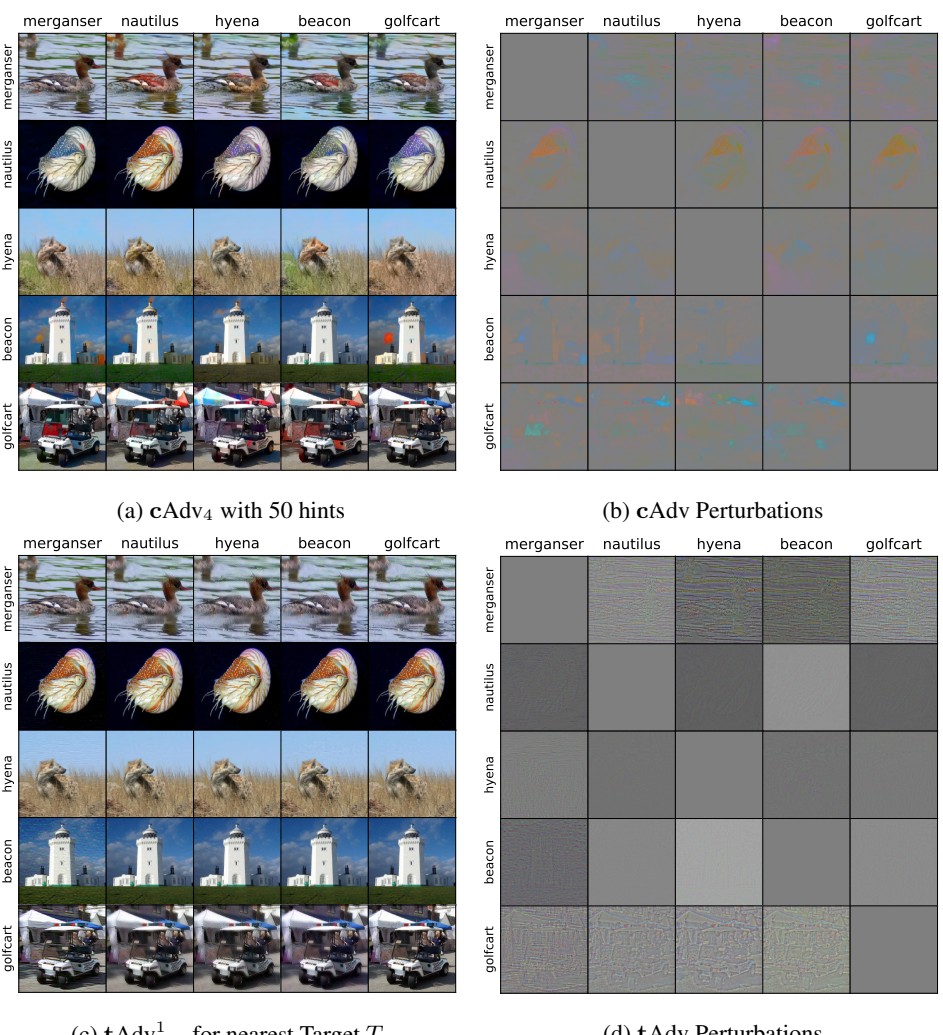

Figure 13: Randomly sampled, semantically manipulated, unrestricted adversarial examples and their perturbations. (a) Adversarial examples generated by **c**Adv attacking Hints and mask with 50 hints. (c) Adversarial examples generated by **t**Adv with $\alpha = 250$, $\beta = 10^{-3}$ and iter= 1 using nearest target $T_s$. (b) and (d) are their respective perturbations. Note that diagonal images are groundtruth images and gray pixels indicate no perturbation.

