# OpenReview forum: "Unrestricted Adversarial Examples via Semantic Manipulation"
_ICLR.cc/2020/Conference — Accept (Poster)_

### Official Review · AnonReviewer3 · 2019-10-14
**Official Blind Review #3**

**Rating:** 6

**Review:**

The paper proposes cAdv and sAdv, two new unrestricted adversarial attack methods that manipulates either color or texture of an image. To these end, the paper employes another parametrized colorization techniques (and texture transfer method) and proposes optimization objectives for finding adversarial examples with respect to each semantic technique. Experimental results show that the proposed methods are more robust on existing defense methods and more transferrable accross models. The paper also performs a user study to show that the generated examples are fairly imperceptible like the C&W attack.

In overall, I agree that seeking a new way of attack is important, and the methods are clearly presented to claim a new message to the community: adversarial examples can be even found by exploiting semantic features that humans also utilize, since DNNs tend to overly-utilize them, e.g. colors. These claims are supported by the experiments showing that the generated examples are more transferrable across robust classifiers. Personally, I liked the idea of using another colorization method to design cAdv and the use of K-means clustering to control the imperceptibility.

- Some readers may wonder how the "averaged-case" corruption robustness behave for both cAdv and sAdv, e.g. considering random colorization. Would it be worse than the robustness on Gaussian noise?
- One of my concerns on tAdv is whether the texture added is indeed effective to reduce the accuracy, or its just from the (yet small) beta term in the objective. Adding an ablation of beta=0 case in the result would much help the understanding of the method.
- Eq 1: I think F should denote the classifier to attack, but the description tells it's the colorization network. As it seems to me that theta is nevertheless for the colorization network, I feel the notation should be refined for better understanding to the readers.

**Experience Assessment:**

I have read many papers in this area.

**Review Assessment: Checking Correctness Of Derivations And Theory:**

N/A

**Review Assessment: Checking Correctness Of Experiments:**

I carefully checked the experiments.

**Review Assessment: Thoroughness In Paper Reading:**

I read the paper at least twice and used my best judgement in assessing the paper.

---

> ### Author Response · Authors · 2019-11-15
> **Response to Reviewer #3**
>
> Thank you for your comments and interest in our work.
>
> Q1: Some readers may wonder how the "averaged-case" corruption robustness behaves for both cAdv and sAdv, e.g. considering random colorization. Would it be worse than the robustness of Gaussian noise?
>
> A1:  Thanks for the interesting question and it’s indeed reasonable to wonder if the classifier is robust against random colorization with the same level of corruption. As we know, classifiers are robust against small random corruptions, whereas large random corruptions might be able to change the classification results but not certainly.  Furthermore, large/unbounded random corruptions, such as random colorization as mentioned, cannot generate semantically aligned images. Therefore, they can be easily spotted by humans as adversarial/abnormal. Our methods are exactly trying to resolve these problems by hiding unbounded adversarial patterns in plain sight.
>
> In particular, some classifiers are not robust towards large random colorizations as seen from the hue and saturation attack [3]. Given sufficiently large perturbation, it is not surprising to see the labels changing. However, hue and saturation attacks give unrealistic images and are not able to do targeted attacks (1.2% on ImageNet). cAdv, on the other hand, has high attack success rates for both targeted and untargeted attacks while staying perceptually realistic. We will clarify this in our revision.
>
> [3] Hosseini, Hossein, and Radha Poovendran. "Semantic adversarial examples." Proceedings of the IEEE Conference on Computer Vision and Pattern Recognition Workshops. 2018.
>
>
> Q2:  One of my concerns on tAdv is whether the texture added is indeed effective to reduce the accuracy, or it’s just from the (yet small) beta term in the objective. Adding an ablation of beta=0 case in the result would much help the understanding of the method.
>
> A2: This is a really interesting and important question w.r.t tAdv. We conducted additional experiments suggested as below. We did an ablation with tAdv varying both beta-term and alpha-term in Table 4 of our Appendix. From our study, it can be inferred that increasing beta does not result in an increase in attack success rate; with a very small beta, the attack is highly successful. However, we did include a row when beta is zero in the submitted version. Here we include the numbers when beta = 0 for two cases when tadv is optimized using LBFGS for one iteration of 14 steps (small texture flow) and 3 iterations of 14 steps (large texture flow) as described in our paper. Note that making beta = 0 does not guarantee that we will be able to reach the target as it is difficult to control texture transfer and stop when the target is reached as small change in texture can make images classify to arbitrary classes. It is also to be noted that beta=0 would be a black box attack as the attack does not have any knowledge about the target classifier.
>
> One iteration of LBFGS with 14 steps of texture transfer
> +--------------------------------------+-----------+-----------+-----------+--------------+
> |                                                  | a = 250 | a = 500 | a = 750 | a = 1000 |
> +--------------------------------------+-----------+-----------+-----------+--------------+
> | Untargeted Attack Success |   25%    |  24.5%  |   25%    |    24.5%   |
> +--------------------------------------+-----------+-----------+-----------+--------------+
> |   Target Attack Success        |   0.5%   |   0.5%   |   0.5%   |     0.5%    |
> +--------------------------------------+-----------+-----------+-----------+--------------+
>
> Three iterations of LBFGS with 14 steps of texture transfer (large texture flow transfer)
> +--------------------------------------+-----------+-----------+-----------+-------------+
> |                                                  | a = 250 | a = 500 | a = 750 | a = 1000 |
> +--------------------------------------+-----------+-----------+-----------+-------------+
> | Untargeted Attack Success |  37.5%  |  40.5%  | 41.5%   |   42.5%   |
> +--------------------------------------+-----------+-----------+-----------+-------------+
> |   Target Attack Success        |   0.5%   |   0.0%   |   0.0%   |     0.0%    |
> +--------------------------------------+-----------+-----------+-----------+-------------+
>
> Both our Table 4 and the above analysis guarantees that tAdv is indeed effective in improving the attack success rate. We will include this analysis in our paper’s supplementary.
>
>
> Q3: I think F should denote the classifier to attack, but the description tells it's the colorization network. As it seems to me that theta is nevertheless for the colorization network, I feel the notation should be refined for better understanding to the readers.
>
> A3: Yes, that’s a typo and you are correct that F is indeed a classifier to attack. We have fixed this in the revision. Sorry about that and thanks for bringing this to our attention.

---

### Official Review · AnonReviewer2 · 2019-10-23
**Official Blind Review #2**

**Rating:** 6

**Review:**

This paper proposed to generate semantically meaningful adversarial examples in terms of color of texture. In order to make manipulated images photo-realistic, colors to be replaced are chosen by energy values, while textures are replaced with style-transfer technique.

The paper is written clearly and organized well to understand. The graphs and equations are properly shown. The idea of using color replacement and texture transfer is interesting and novel.

A somewhat weakness is that the discriminator - a pretrained ResNet 50 - is too weak for this scenario. What about a ResNet 50 trained on augmented datasets with color jittering?
What about finetuned ResNet 50 with taking color channel as explicit input, since the attack uses this additional info.

As tAdv attack seems to manipulate high frequency texture of images, how about applying a Gaussian filter on the images and feed into the discrimimator again? Is that attack still effective or not?

**Experience Assessment:**

I have read many papers in this area.

**Review Assessment: Checking Correctness Of Derivations And Theory:**

I assessed the sensibility of the derivations and theory.

**Review Assessment: Checking Correctness Of Experiments:**

I assessed the sensibility of the experiments.

**Review Assessment: Thoroughness In Paper Reading:**

I read the paper thoroughly.

---

> ### Author Response · Authors · 2019-11-15
> **Response to Reviewer #2**
>
> Thanks for the interesting question and suggestions.
> Q1: A somewhat weakness is that the discriminator - a pre-trained ResNet 50 - is too weak for this scenario. What about a ResNet 50 trained on augmented datasets with color jittering? What about finetuned ResNet 50 with taking color channel as explicit input, since the attack uses this additional info.
>
> A1:  We conducted extra two sets of experiments as you described. First, we finetune a ResNet50 model for 35 epochs using:
>
> a) data augmented with brightness, contrast, hue, and saturation jittering
> Top1 accuracy: 75.842%
> b)  LAB colorspace as input
> Top1 accuracy: 76.206
>
> We then apply cAdv to both models. We randomly select 60 images for 3 different classes and use targeted cAdv to attack the two models to two random different classes. The final accuracies of the two models on the cAdv images is as follows
>
> Top1 accuracy for a): 0.000%
> Top1 accuracy for b): 0.000%
>
> Even when trained with data augmentation or taking in LAB colorspace, cAdv is able to easily attack the two models.
>
>
> Q2: As tAdv attack seems to manipulate the high-frequency texture of images, how about applying a Gaussian filter on the images and feed into the discriminator again? Is that attack still effective or not?
>
> A2: Thanks again for the interesting question, and on the suggested experiments regarding  tAdv: In Table 2, under Feature Squeezing, we reported scores for two types of Gaussian Filtering -- 2x2, 3x3. tAdv outperforms other baselines significantly and is a stronger attack on Gaussian Filtering. Our intuition for why tAdv is stronger despite having high-frequency perturbations is because of their structured pattern and are not local like most of the recent other attacks and are therefore able to pass through filtering operations. We will make this clear in our Table as well as the caption in our revision and sorry for the confusion.

---

### Official Review · AnonReviewer1 · 2019-10-23
**Official Blind Review #1**

**Rating:** 6

**Review:**

This paper introduces two new adversarial attacks: one is generating adversarial examples by colouring the original images and the other is by changing textures of the original images. Specifically, the former one minimises the cross-entropy between the output of the classifier and the target label with the network weights of a pre-trained colourisation network. While the latter minimises the cross-entropy as well as the loss that defines the texture differences.

I think the general idea of going beyond perturbations of pixel values in this paper is interesting and the proposed approaches of attacking on colour and textures are intuitive and reasonable. The results seem to be promising with comprehensive experiments including whitebox attack, blackbox attack by transferring, and attacks on defences.

The paper overall is well-written and easy to follow. But I think the part of attacking for captioning is a bit distracted and there is no comparison with others on this task. I expect existing attacks on pixel can also do this task.

**Experience Assessment:**

I have read many papers in this area.

**Review Assessment: Checking Correctness Of Derivations And Theory:**

I assessed the sensibility of the derivations and theory.

**Review Assessment: Checking Correctness Of Experiments:**

I assessed the sensibility of the experiments.

**Review Assessment: Thoroughness In Paper Reading:**

I read the paper at least twice and used my best judgement in assessing the paper.

---

> ### Author Response · Authors · 2019-11-15
> **Response to Reviewer #1**
>
> Thank you for your comments and interest in our work.
> Q1:  The paper overall is well-written and easy to follow. But I think the part of attacking for captioning is a bit distracted and there is no comparison with others on this task. I expect existing attacks on pixels can also do this task.
>
> A1: We agree the captioning section was oddly arranged in our current submission. We will move the section towards the end of the paper to show it as another attack application.
> There are some pixel-level attacks against image captioning models [1,2] , but these captioning models are different so it is hard to directly compare. We will add corresponding discussion in our related work.
>
>
> [1] Yan Xu, Baoyuan Wu, Fumin Shen, Yanbo Fan, Yong Zhang, Heng Tao Shen, and Wei Liu. Exact adversarial
> attack to image captioning via structured output learning with latent variables. In Proceedings of the IEEE
> Conference on Computer Vision and Pattern Recognition, pp. 4135–4144, 2019.
> [2] Chen, H., Zhang, H., Chen, P.Y., Yi, J. and Hsieh, C.J., 2017. Attacking visual language grounding with adversarial examples: A case study on neural image captioning. 56th Annual Meeting of the Association for Computational Linguistics (ACL 2018)

---

### Decision · Program_Chairs · 2019-12-19

**Decision:**

Accept (Poster)

**Comment:**

In this paper, the authors present adversarial attacks by semantic manipulations, i.e., manipulating specific detectors that result in imperceptible changes in the picture, such as changing texture and color, but without affecting their naturalness. Moreover, these tasks are done on two large scale datasets (ImageNet and MSCOCO) and two visual tasks (classification and captioning). Finally, they also test their adversarial examples against a couple of defense mechanisms and how their transferability. Overall, all reviewers agreed this is an interesting work and well executed, complete with experiments and analyses. I agree with the reviewers in the assessment. I think this is an interesting study that moves us beyond restricted pixel perturbations and overall would be interesting to see what other detectors could be used to generate these type of semantic manipulations. I recommend acceptance of this paper.